Single-view 3D reconstruction via dual attention

http://orcid.org/0009-0006-0363-2240 Li Chenghuan 1
Xiao Meihua 1 xiaomh@ecjtu.edu.cn
Li Zehuan 1
Chen Fangping 2
Wang Dingli 1
1 Software of School, East China JiaoTong University , Nanchang, JiangXi , China
2 Jiangxi University of Software Professional Technology , Nanchang, JiangXi , China
Balas Valentina Emilia
Electronic publication date: 2024 Oct 22
Publication date: 2024
Volume: 10
Electronic Location ID: e2403
Received 2024 May 2; Accepted 2024 Sep 19
Copyright: © 2024 Li et al.
Copyright year: 2024
Copyright holder: Li et al.
License: This is an open access article distributed under the terms of the Creative Commons Attribution License, which permits unrestricted use, distribution, reproduction and adaptation in any medium and for any purpose provided that it is properly attributed. For attribution, the original author(s), title, publication source (PeerJ Computer Science) and either DOI or URL of the article must be cited.
License URL: https://creativecommons.org/licenses/by/4.0/

Keywords: 3D reconstruction, Computer vision, Deep learning, Transformer, Selective state space model, Voxel model

Funding: National Natural Science Foundation of China 62362033 and 61962020 Double Thousand Talent Plan of Jiangxi Province jxsq2023201009 Natural Science Foundation of Jiangxi Province 20224ACB202006 This research was funded by the National Natural Science Foundation of China, grant number 62362033, 61962020, the Double Thousand Talent Plan of Jiangxi Province (No. jxsq2023201009), and the Natural Science Foundation of Jiangxi Province (No. 20224ACB202006). The funders had no role in study design, data collection and analysis, decision to publish, or preparation of the manuscript.

==============================
Constructing global context information and local fine-grained information simultaneously is extremely important for single-view 3D reconstruction. In this study, we propose a network that uses spatial dimension attention and channel dimension attention for single-view 3D reconstruction, named R3Davit. Specifically, R3Davit consists of an encoder and a decoder, where the encoder comes from the Davit backbone network. Different from the previous transformer backbone network, Davit focuses on spatial and channel dimensions, fully constructing global context information and local fine-grained information while maintaining linear complexity. To effectively learn features from dual attention and maintain the overall inference speed of the network, we do not use a self-attention layer in the decoder but design a decoder with a nonlinear reinforcement block, a selective state space model block, and an up-sampling Residual Block. The nonlinear enhancement block is used to enhance the nonlinear learning ability of the network. The Selective State Space Model Block replaces the role of the self-attention layer and maintains linear complexity. The up-sampling Residual Block converts voxel features into a voxel model while retaining the voxels of this layer. Features are used in the up-sampling block of the next layer. Experiments on the synthetic dataset ShapeNet and ShapeNetChairRFC with random background show that our method outperforms recent state of the art (SOTA) methods, we lead by 1% and 2% in IOU and F1 scores, respectively. Simultaneously, experiments on the real-world dataset Pix3d fully prove the robustness of our method. The code will be available at https://github.com/epicgzs1112/R3Davit.

Introduction

Single-view 3D reconstruction involves restoring the shape of an object based on a single-view image of the object. This is a very challenging research topic that involves the fields of computer vision and computer graphics. One of the most important challenges is how to extract features from a single image to generate corresponding 3D objects. Currently, researchers have provided three solutions, including convolutional neural network (CNN)–based methods (Xie et al., 2020; Zhu et al., 2023b), recurrent neural network (RNN)-based methods (Choy et al., 2016; Kar, Häne & Malik, 2017), and transformer-based methods (Wang et al., 2021; Yagubbayli et al., 2021; Shi et al., 2021; Zhu et al., 2023a; Yang et al., 2023). In this study, we focus on Transformer-based single-view 3D reconstruction using voxel models as expressions.

The Vision Transformer (Alexey, 2020) segments the image into a series of fixed-size nonoverlapping patches, which are then flattened and fed into the Transformer model as a sequence. Each patch is considered a position in the sequence. Then, through the Transformer’s attention mechanism, the model can learn the relationship between different locations in the image, thereby capturing global contextual information. However, Vision Transformer relies heavily on large datasets, fixed feature map size loss, and multi-scale information, and there is no connection between patches. At the same time, using patches as sequences still does not escape the curse of trainable sequence length in the field of NLP. Therefore, Transformer-based 3D reconstruction methods (Wang et al., 2021; Yagubbayli et al., 2021; Shi et al., 2021) perform poorly in single-view reconstruction.

Recently, the distillation-based Vision Transformer (Touvron et al., 2021) and Hierarchical Vision Transformer (Liu et al., 2021) have shown great promise in the field of computer vision. The distillation-based Vision Transformer relies on distilled tokens to ensure that students learn from the teacher through attention, usually from the convolutional network teacher, overcoming the shortcomings of the traditional Vision Transformer’s reliance on large data sets. The layer-based Transformer combines the advantages of CNN and Transformer. On the one hand, it has the advantages of CNN in processing large-size images due to the local attention mechanism. On the other hand, it has the advantage of Transformer in that it can model long-range dependencies through a shifted window scheme. However, they all only perform attention calculations in the spatial dimension of the image and ignore the global context information of the channel dimension.

In this article, we propose a novel simple 3D reconstruction model, namely R3Davit, based on Davit (Ding et al., 2022). More specifically, R3Davit consists of two modules: the backbone network Davit (Ding et al., 2022) and a decoder with a nonlinear reinforcement block, an SSM block, and an up-sampling residual block.

The contributions can be summarized as follows: (1) We find that performing attention calculations in both spatial and channel dimensions simultaneously is very effective for 3D reconstruction. (2) To fully learn features from dual-dimensional attention without inserting self-attention blocks, we design a simple and effective decoder. (3) Experiments on the synthetic dataset ShapeNet (Chang et al., 2015) and ShapeNetChairRFC with random background show that our method outperforms all previous SOTA methods. Simultaneously, experiments on the real-world dataset Pix3d (Sun et al., 2018) fully prove the robustness of our method.

Related work

Since the release of the 3D reconstruction datasets ShapeNet (Chang et al., 2015) and Pix3d (Sun et al., 2018), the research interest in 3D reconstruction based on deep learning has further increased. According to the expression method of the model, it can be divided into point cloud 3D reconstruction (Jia et al., 2020; Fan, Su & Guibas, 2017), voxel 3D reconstruction (Xie et al., 2020; Zhu et al., 2023b; Choy et al., 2016; Kar, Häne & Malik, 2017; Wang et al., 2021; Yagubbayli et al., 2021; Shi et al., 2021; Zhu et al., 2023a; Yang et al., 2023), polygon 3D reconstruction (Shen et al., 2021; Laine et al., 2020; Munkberg et al., 2022; Wang et al., 2018; Wen et al., 2019), and implicit 3D reconstruction represented by NERF (Mildenhall et al., 2021; Barron et al., 2021; Deng et al., 2022) and 3DGS (Kerbl et al., 2023; Qin et al., 2023) and others.

In this work, we focus on voxel-based 3D reconstruction. Choy et al. (2016) designed a three-dimensional long short term memory (LSTM) network based on the LSTM network to process the encoded information of a single image. The network consists of CNN, 3D-LSTM, and 3D-CNN. CNN encodes the image into low-dimensional features and sends them to 3D. LSTM updates the latent encoding and finally uses 3DCNN decoding, which uses the sum of voxel cross entropy as the loss function to train the network, reconstruct the voxel model, and achieve end-to-end reconstruction of the 3D model for a single image. Due to long-term memory loss, RNN forgets important features of earlier input images. The Pix2vox++ network proposed by Xie et al. (2020) uses a residual network as the backbone network and consists of an encoder, decoder, fusion device, and refiner. Although this network has achieved encouraging results in single-view reconstruction, more modules have greatly increased the training parameters and GPU memory usage. Shi et al. (2021) proposed 3D-RETR to fill the gap of whether transformer can be used for 3D reconstruction. It used a pre-trained transformer to extract visual features from 2D input images. Another transformer decoder is then used to obtain voxel features. Finally, the CNN decoder takes the voxel features as input to obtain the reconstructed object. Because the encoder inherits from the vision transformer, it also retains the shortcoming of the vision transformer, i.e., it cannot establish long-range correlation and multi-scale features of the view. Zhu et al. (2023b) proposed a global-aware attention-based fusion method that established the correlation between each branch and the global world. Although it achieved good results under multi-view input, the performance dropped significantly in single-view mode. Zhu et al. (2023a) designs an inter-view-decoupled block to mine the correlation between similar patches from different views. Through token clustering, tokens from all branches are compressed into a fixed-size compact representation. But ignores the similarity between dissimilar Tokens.

Method

In this section, we introduce the spatial and channel dimensions of the backbone network and the specific details of the decoder. The overall framework of our proposed method is illustrated in Fig. 1.

Figure 1 Our proposed R3Davit overall structure.

Encoder

The encoder is based on Davit (Ding et al., 2022) and consists of four stages. Taking the Davit (Ding et al., 2022) base version as an example, Each stage contains 1, 1, 9, and 1 dual transformer block. Each dual transformer block contains spatial window self-attention and channel group self-attention. The attention calculation layer is the layernorm layer and the feedforward neural network. Each dual transformer block is preceded by a patch embedding layer. The details of the dual transformer block are illustrated in Fig. 2.

Figure 2 Each block has two attention calculation layers.

The former layer performs spatial window self-attention calculations to establish local fine-grained information, and the latter performs channel group self-attention computation to construct global context information.

We denote the input image of an object from one view as x∈RH×W×C, H, W, C represent the height, width, and channels of the image, respectively. The patch embedding layer consists of a 2D convolution layer with out channel = D, kernel size =7, stride =4, padding =3, and a layernorm layer. The patch embedding layer divides the image into nonoverlapping patches, each patch contains 4×4 pixels of information. After the patch embedding layer, the image features enter the dual transformer block for attention calculation. The feature map dimensions change at each stage, as illustrated in Table 1.

Table 1 Dimensions of feature maps at each stage of the encoder.

Stage	Dimension	
0	224×224×3	
1	3,136×128	
2	784×256	
3	196×512	
4	49×1,024	

Spatial window attention

Spatial Window Attention works like other Vision Transformer (Alexey, 2020) work, performing multi-head self-attention directly in local windows. We denote the windows as Nw and each window consists of Pw patches. Then, window attention can be represented by:

(1) Awindow(Q,K,V)={A(Qi,Ki,Vi)}i=0Nw

where Qi,Ki,Vi Vi∈RPw×Ch are local window queries, keys, and values.

Channel group attention

In contrast to the previous Transformer backbone network (Alexey, 2020; Liu et al., 2021; Touvron et al., 2021), Davit not only performs attention from the pixel patch level but also from the channel patch level. The channel patch can be obtained simply by transposing the pixel patch. To maintain linear complexity, Davit groups channels and performs self-attention on each group separately. We denote the number of groups as Ng and each group consists of Cg channel. Then, channel group attention can be represented by:

(2) Achannel(Q,K,V)={Agroup(Qi,Ki,Vi)T}i=0NgAgroup(Qi,Ki,Vi)=softmax[QiTKiCg]ViT

where Qi,Ki,Vi Vi∈RP×Cg are grouped channel-wise image-level queries, keys, and values.

Decoder

In this section, we introduce the design of the decoder in detail. The decoder consists of a nonlinear reinforcement block, a selective state space model block, and an up sample block. The nonlinear reinforcement block enhances the nonlinear learning ability of the decoder. The selective state space model block is used to replace the traditional Transformer in establishing long-range dependencies. The Up Sample layer samples the voxel features and finally becomes a voxel model through the Sigmod activation function. Details of the Decoder are illustrated in Fig. 3.

Figure 3 An illustration of the decoder.

Nonlinear reinforcement block

Inspired by Tran et al. (2018), the complete 3D convolution can be decomposed into two-dimensional and one-dimensional convolution, and the decomposition method can have one more RELU activation layer compared with using a single 3D convolution, thus increasing the nonlinear learning ability of the network. Two-dimensional and one-dimensional convolution can overcome the shortcomings of the large number of 3D convolution parameters. Therefore, we use R(2+1)D (Tran et al., 2018) convolution to construct our nonlinear reinforcement block and design it in the residual connection framework. The nonlinear reinforcement block consists of three R(2+1)D (Tran et al., 2018) convolutions, and a RELU activation function is inserted between each R(2+1)D (Tran et al., 2018) convolution. Details about the nonlinear reinforcement block as illustrated in Fig. 4.

Figure 4 Nonlinear reinforcement block.

Selective state space model block

Previous work (Shi et al., 2021; Zhu et al., 2023a; Yang et al., 2023) often uses traditional Transformer layers to further build long-range dependencies in the design of decoders. However, because the length of the token sequence received by the decoder is too long, the training parameters increase exponentially, resulting in slow model inference. In contrast to previous work, inspired by Gu & Dao (2023) we use SSM block to replace the traditional Transformer layer to maintain linear complexity while building long-range dependencies. the selective state space model (Gu & Dao, 2023) is a type of sequence model used in deep learning that maps a one-dimensional function or sequence x(t)∈R→y(t)∈R through implicit potential states h(t)∈RN. Concretely, The selective state space model (Gu & Dao, 2023) are defined with four parameters (Δ,A,B,C), which define a sequence-to-sequence transformation in two stages: discretization and computation.

The Selective State Space Model could be formulated as:

(3) h′(t)=Ah(t)+Bx(t)y(t)=Ch(t)

the discretization could be formulated as:

(4) ht=A¯ht−1+B¯xtyt=ChtK¯=(CB¯,CAB¯,...,CA¯kB¯)y=x∗K¯.

The first stage transforms the continuous parameters (Δ,A,B,C) to “discrete parameters” (A¯,B¯) through fixed formulas A¯=fA(Δ,A) and B¯=fB(Δ,A,B), where the pair (fA,fB) is called a discretization rule. Various rules can be used such as the zero-order hold (ZOH) defined in Eq. (5).

(5) A¯=exp(ΔA)B¯=(ΔA)−1(exp(ΔA)−I)⋅ΔB.

After the parameters have been transformed from (Δ,A,B,C)→(A¯,B¯,C), the model can be computed in two ways, either as a linear recurrence or a global convolution. Detail about selective state space model block as illustrated in Fig. 5.

Figure 5 Selective state space model block.

Up sample block

The up sampling block consists of three 3D transposed convolutional layers, each of which is inserted into a BatchNorm3D layer and a RELU nonlinear activation function. Each 3D transposed convolutional layer kernel size = 4, stride = 2, and padding = 1. Details of the up sample block are illustrated in Fig. 6.

Figure 6 Up sample block.

Loss function

Following 3D-RETR (Shi et al., 2021), we use Dice loss (Milletari, Navab & Ahmadi, 2016) as the loss function. The Dice loss could be formulated as:

(6) ι=1−∑i=1323riti∑i=1323ri+ti−∑i=1323(1−ri)(1−ti)∑i=13232−ri−ti

where ri represent the confidence of i-th voxel grid on the reconstructed volume, and ti represent the confidence of i-th voxel grid on the ground truth.

Experiments

We used an A4000 GPU with 16 GB of memory. Training took 2 days, depending on the exact setting. The batch size was set to 16 for all experiments.

Evaluation metric

We use IOU and F-score 1% to measure the performance of the model. The higher the value, the better the result.

The mean Intersection-over-Union (IOU) is formulated as:

(7) IoU=∑(i,j,k)I(r^(i,j,k)>t)I(r(i,j,k))∑(i,j,k)I[I(r^(i,j,k)>t)+I(r(i,j,k))]

where r^(i,j,k) represent the predicted occupancy probability and r(i,j,k) represent the ground truth at (i,j,k). t denotes a voxelization threshold.

The F-score 1% is formulated as:

(8) Fscore(d)=2P(d)R(d)P(d)+R(d)

where P(d) and R(d) denote the precision and recall for a distance threshold between prediction and ground truth. F-score 1% indicates the F-score value when d is set to 0.01.

Datasets

Following previous work (Choy et al., 2016; Wen et al., 2019; Yagubbayli et al., 2021; Zhu et al., 2023a), we use a subset of the ShapeNet (Chang et al., 2015) dataset to train the network and a subset of the Chair class in the ShapeNet dataset and a background randomly sampled from the Sun data were synthesized into the ShapeNetChairRFC dataset, and the model trained on ShapeNetChairRFC was used to verify the Pix3d (Sun et al., 2018) dataset.

Results

Results on ShapeNet

For single-view 3D reconstruction on ShapeNet (Chang et al., 2015), we compared our results with recent state-of-the-art models, including Pix2vox++ (Xie et al., 2020), 3D-RETR (Shi et al., 2021), and Umiformer (Zhu et al., 2023a). Table 2 show the IOU and F1 score results, and more importantly, the visualization results Table 3. We can observe from Table 3 that our method outperforms all previous models in terms of overall IOU and F1 scores. Furthermore, except for IOU shown, our method outperforms all other methods in 10 out of 13 categories, while it can be observed from the visualization results Table 2 that we achieve the best in the prediction of chair leg joints result, The most important is the Overall value, where our work leads by 1% in the IOU metric and 2% in the F1 metric compared to UMIFormer.

Table 2 Compare to recent SOTA work, and IOU/F1.

Bold indicates the best performance.

Category	Pix2vox++	3D-RETR	UMIFormer	R3Davit (Ours)	
Airplane	0.673/0.549	0.705/0.593	0.701/0.581	0.726/0.613	
Bench	0.607/0.445	0.654/0.498	0.644/0.483	0.674/0.518	
Cabinet	0.798/0.405	0.808/0.422	0.818/0.435	0.826/0.451	
Car	0.857/0.541	0.859/0.548	0.864/0.555	0.873/0.576	
Chair	0.581/0.286	0.589/0.292	0.609/0.305	0.623/0.323	
Display	0.548/0.285	0.566/0.290	0.610/0.337	0.603/0.331	
Lamp	0.456/0.319	0.478/0.328	0.510/0.357	0.509/0.361	
Speaker	0.720/0.282	0.727/0.302	0.755/0.318	0.748/0.314	
Rifle	0.617/0.547	0.671/0.606	0.674/0.606	0.684/0.618	
Sofa	0.724/0.375	0.736/0.387	0.747/0.4001	0.761/0.421	
Table	0.619/0.379	0.626/0.387	0.662/0.416	0.668/0.427	
Telephone	0.809/0.613	0.768/0.542	0.809/0.600	0.835/0.641	
Watercraft	0.602/0.383	0.636/0.418	0.645/0.424	0.656/0.444	
Overall	0.670/0.417	0.679/0.432	0.700/0.447	0.711/0.464	

Table 3 Visualization results on ShapeNet.

Pix2vox++	3D-RETR	UMIFormer	R3Davit (Ours)	GT	
					
					
					
					

Results on ShapeNetChairRFC

Following previous work (Xie et al., 2020; Zhu et al., 2023a), We sampled randomly from the Sun dataset (Xiao et al., 2010) for background and ShapeNet Chair class subset to form a new dataset, ShapeNetChairRFC. ShapeNetChairRFC with random background datasets was used to verify the network robustness on the Pix3d (Sun et al., 2018) dataset. We can observe from Table 4 that our method still outperforms all previous models in terms of overall IoU and F1 scores on ShapeNetChairRFC dataset.

Table 4 Results on ShapeNetChairRFC dataset.

Bold indicates the best performance.

Pix2vox++
0.359/0.126	3D-RETR
0.542/0.254	UMIFormer
0.570/0.268	R3Davit (ours)
0.588/0.290	GT	
					
					

Results on Pix3D

It is very important to use the Pix3D (Sun et al., 2018) dataset to evaluate the generalization performance of the model. We follow Legoformer (Yagubbayli et al., 2021) and use a mask to segment out the background and replace it with a constant color. The resulting images are input to R3Davit trained on Shapenet (Chang et al., 2015), and we report examples of predictive models in Table 5.

Table 5 Visualization results on Pix3d.

Pix2vox++	UMIFormer	R3Davit (Ours)	GT	
				
				

Ablation study

Encoder

Similar to recent Vision Transformer-based methods (Shi et al., 2021; Zhu et al., 2023a), they used Vision Transformer’s pre-trained model on Imagenet (Deng et al., 2009), whereas we used Davit’s (Ding et al., 2022) pre-trained model on Imagenet (Deng et al., 2009). Not using the pre-trained Davit’s (Ding et al., 2022) encoder in experiments will result in a significant drop in model performance.

Decoder

To comprehensively evaluate our method and other SOTA methods, we further compared the network training parameter size and evaluation indicators in Table 5, For fair comparison we removed the merger from other studies. It can be clearly observed that our network achieves higher metrics despite having fewer training parameters. Simultaneously, to prove that our proposed decoder is more effective, We substituted the decoder used in other studies, result in Table 6.

Table 6 Comparison of the parameter size and accuracy.

Bold indicates the best performance.

Module	Pix2vox++	3D-RETR	UMIFormer	R3Davit (ours)	
Encoder	5.5 M	85.8 M	88.1 M	85.5 M	
Decoder	55.8 M	77.3 M	76.9 M	14.5 M	
Refiner	34.8 M	∖	∖	∖	
Sum	96.1 M	163.1 M	165 M	100 M	
Overall	0.670/0.417	0.679/0.432	0.700/0.447	0.711/0.464	

In order to validate the efficacy of our proposed nonlinear reinforcement block and selective state space model block, we have developed a baseline network that does not include these components. This network has been trained on both the ShapeNet (Chang et al., 2015) dataset, result in Table 7.

Table 7 Ablation study about Decoder, NRB (nonlinear reinforcement block), SSSMB (selective state space model block).

Bold indicates the best performance.

	NRB	SSSMB	ResConnect	IOU/F1	
Baseline				0.709/0.456	
Baseline+NRB	✓			0.710/0.461	
Baseline+NRB+SSSMB	✓	✓		0.711/0.462	
Full	✓	✓	✓	0.711/0.464	

Conclusion and limitations

In this work, we propose a novel network based on Davit (Ding et al., 2022) for single-view 3D reconstruction, achieving SOTA accuracy on both ShapeNet (Chang et al., 2015) and Pix3d (Sun et al., 2018) datasets. The encoder extracts image features from single-view data using spatial dimension attention and channel dimension attention, simultaneously constructing global context information and local fine-grained information. To effectively learn features from dual attention and maintain the overall inference speed of the network, we do not use a self-attention layer in the decoder but design a decoder with a nonlinear reinforcement block, a selective state space model block, and an upsampling residual block.

Supplemental Information

Supplemental Information 1 Results from 3dretr.

Supplemental Information 2 Results from pix2vox++.

Supplemental Information 3 GroundTruth.

Supplemental Information 4 Results from our method.

Supplemental Information 5 Results from uniformer.

Additional Information and Declarations

Competing Interests

Author Contributions

Data Availability

The authors declare that they have no competing interests.

Chenghuan Li conceived and designed the experiments, performed the experiments, performed the computation work, prepared figures and/or tables, and approved the final draft.

Meihua Xiao conceived and designed the experiments, authored or reviewed drafts of the article, and approved the final draft.

Zehuan Li performed the experiments, prepared figures and/or tables, and approved the final draft.

Fangping Chen analyzed the data, prepared figures and/or tables, and approved the final draft.

Dingli Wang performed the experiments, analyzed the data, prepared figures and/or tables, and approved the final draft.

The following information was supplied regarding data availability:

The code is available at GitHub and Zenodo:

- https://github.com/epicgzs1112/R3Davit

- li, chenghuan. (2024). ShapeChairRFC dataset (Version 1) [Data set]. Zenodo. https://doi.org/10.5281/zenodo.13294314.

The ShapeNet dataset from Princeton University and Stanford University is available at: https://shapenet.org.

The ShapeNetChairRFC dataset is available at Zenodo: li, chenghuan. (2024). ShapeChairRFC dataset (Version 1) [Data set]. Zenodo. https://doi.org/10.5281/zenodo.13294314.

The Pix3d dataset is available at: http://pix3d.csail.mit.edu.

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
