# Peer review of "Single-view 3D reconstruction via dual attention"

_PeerJ Computer Science, doi:10.7717/peerj-cs.2403_

## Round 0.1 · original submission · Minor Revisions

The authors must introduce the corrections suggested by reviewers.

Reviewer 1 ·

Basic reporting

The paper introduces a novel 3D reconstruction method that utilizes a dual transformer-based encoder and a decoder with a Selective State Space Model (SSM) block. The encoder is based on the Davit model, which consists of 4 stages with dual transformer blocks. Each block has a spatial window self-attention layer and a channel group self-attention layer to capture both local and global features. The decoder consists of three main components:Nonlinear Reinforcement Block which uses R(2+1)D convolutions to enhance the nonlinear learning ability; Selective State Space Model (SSM) Block which replaces the traditional Transformer layer to maintain linear complexity while building long-range dependencies; and the up Sample Block.

The method presented in the paper is somewhat novel, and the paper also validates the effectiveness of the method through experiments. Compared to the SOTA methods, the performance of this paper is overall superior to them. However, the layout of the figures in this paper needs improvement to further enhance its quality. Please change Figures 4, 5, and 6 to a horizontal layout to save space and make them easier to read and understand.

Experimental design

The paper compares the state-of-the-art (SOTA) methods published after 2020 on the ShapeNet, ShapeNetChairRFC, and Pix3D datasets, and presents the best performances in terms of IOU and F1 score. The ablation study also effectively demonstrated the effectiveness of each component. But I got confused by the sentence in section 3.3.3 "The resulting images are input to R3D-SWIN". What is R3D-SWIN here?

Validity of the findings

The source code is available at https://github.com/epicgzs1112/R3Davit. The authors are confident in their work. The paper mentions multi-view reconstruction in the conclusion, but it is not the focus of this study. The authors do not need to concern themselves with this field.

Cite this review as
Anonymous Reviewer (2024) Peer Review #1 of "Single-view 3D reconstruction via dual attention (v0.1)". PeerJ Computer Science

Reviewer 2 ·

Basic reporting

The manuscript is mostly well written, making it (overall) accessible to a wide range of readers in the field of computer vision and 3D reconstruction. The terminology used is appropriate and well-defined, ensuring that readers can follow the technical details without ambiguity. Nevertheless, there are specific issues with some paragraphs difficult to grasp. For example:
1st paragraph of 3.2
1st paragraph of 3.3.2
1st paragraph of 4.1
There are also some typography issues, like capital letters within paragraphs, missing spacings after commas, etc. Also some names, like the names of datasets, should be presented with capital first letter.

The introduction and background sections provide an adequate context for the study, clearly explaining the challenges and current approaches in single-view 3D reconstruction. The literature review is thorough and covers a range of methods, including CNN-based, RNN-based, and transformer-based approaches. This thorough review justifies the focus on transformer-based methods and highlights the gap that the current study aims to fill.

Figures and tables included in the manuscript are relevant and well-labeled. They effectively illustrate the experimental setup, the architecture of the proposed method, and the results of the study. Each figure and table is accompanied by detailed captions that enhance understanding. In section 3.3.1 the correct reference for the visualisation of results would be Table 3 instead of Table 2.

Experimental design

The research question is well-defined, focusing on improving single-view 3D reconstruction by leveraging dual attention mechanisms. The study fits well within the scope of the journal, addressing a significant gap in current research and providing novel contributions to the field.

The methodology is described in sufficient detail to enable replication. The design of the encoder and decoder, the integration of dual attention mechanisms, and the novel components such as the nonlinear reinforcement block, selective state space model block, and up-sampling residual block are explained clearly. The use of both synthetic and real-world datasets adds robustness to the experimental design.

The experimental setup is appropriate and includes a comprehensive set of datasets (ShapeNet, ShapeNetChairRFC, Pix3d). The choice of metrics (Intersection over Union (IoU) and F1 scores) for evaluating performance is justified, as they are standard and widely accepted in the field.

The detailed description of the architecture and the training process, including hyperparameters and data preprocessing steps, ensures that other researchers can replicate the study. The code and datasets used should be made available to the community to facilitate this replication.

Validity of the findings

The findings are robust and supported by experimental data. The results show that the proposed method outperforms recent state-of-the-art methods in single-view 3D reconstruction. The improvements in IoU and F1 scores are statistically important, demonstrating the effectiveness of the dual attention mechanism.

The paper provides a thorough analysis of the results, including an ablation study, which strengthens the validity of the findings. The robustness of the method is further validated on real-world datasets, adding practical relevance to the study.

Cite this review as
Anonymous Reviewer (2024) Peer Review #2 of "Single-view 3D reconstruction via dual attention (v0.1)". PeerJ Computer Science

---

## Round 0.2 · Minor Revisions

There are some few typos to correct.

So, minor revisions are needed before acceptance.

Reviewer 1 ·

Basic reporting

This is the paper I reviewed two months ago, and some of my concerns were posed in the previous round of reviewing.
I think it can be accepted now.

Experimental design

The experiment is sufficient and convicing.

Validity of the findings

The ablation study enhances the validity of the findings. The robustness of the method is confirmed on real-world datasets, adding practical significance to the research.

Cite this review as
Anonymous Reviewer (2024) Peer Review #1 of "Single-view 3D reconstruction via dual attention (v0.2)". PeerJ Computer Science

Reviewer 2 ·

Basic reporting

The authors have made an effort to address the concerns of the reviewer, nevertheless there are still issues with the text that significantly lower its quality and readability.

There is a major issue in typography: for example all fullstops, commas, parentheses, etc that require a space before them, seem to be treated without the proper "respect"...

The format of using the references changes from point to point and seems to be off...for example, Line 33: Xie et al. (2019) should probably be (Xie et al., 2019).

Furthermore, Line 152: “S4 models”: this needs some explanation AND reference(s).
There are also consistency issues, with some words appearing other times with capital and other with lowercase letters, for example:
- Line 68: “shapenet” and “pix3d”
- Legend Table 3: “shapenet”
- Legend Table 5: “pix3d”
- Line 161: “Fig.5.” should probably be "Figure 5"
- “3dretr“ and “3d-retr” —> 3D-RETR
- “vit” —> ViT. Add also an explanation. For example, in Line 37: “The Vision Transformer” —> “The Vision Transformer (ViT)” OR Line 116
- “davit” —> “Davit”

In addition, many figures are captioned as Tables, because probably the authors used table formatting to create the appropriate space and present the graphs clearly. Nevertheless, these are still Figures and note Tables.

Some more issues:
- Legend Table 4: “Result of ShapeNetChairRFC dataset.” —> “Results on..”
- Legend Table 5: “Result” —> “Results”
- Line 160-161: “Detail about Selective State Space Model 161  Block as illustrated in Fig.5. “
- Line 155: “, the discretization...”
- Line 184: "Following previous work" should be more specific, mentioning the particular studies being followed.
- Line 162: consider rephrasing "The upper adoption block consists of three 3D transposed convolutional layers..." —> "The up-sampling block consists of three 3D transposed convolutional layers..." for clarity.
- Legend Table 2: “Without bells and whistles” —> maybe remove such expressions... And also rephrase the legend to be grammatically correct.
- Line 77: “In this work,we focus on voxel-based 3D reconstruction. “ —> add this sentence to the next paragraph.
- Section 2.1 “Encoder”, the first paragraph needs refinement.
- Legend Table 7. What does “UMIFormer* use our decoder” mean?
- Conclusion refinement: Additionally, suggest possible avenues for future research, such as exploring the use of your method in other datasets or extending it to multi-view reconstruction tasks.
- Mention any qualitative differences observed in the reconstructions.

Experimental design

Ok

Validity of the findings

Ok

Cite this review as
Anonymous Reviewer (2024) Peer Review #2 of "Single-view 3D reconstruction via dual attention (v0.2)". PeerJ Computer Science

---

## Round 0.3 · accepted · Accept

The paper was very well improved and can be accepted.

Reviewer 1 ·

Basic reporting

this is the third version of this paper. I can be accepted now.

For future research, it's ok to extend your work to multi-view reconstruction.

Experimental design

The experiment is sufficient and convincing.

Validity of the findings

The ablation study enhances the validity of the findings. The robustness of the method is further confirmed on real-world datasets, adding practical significance to the research.

Cite this review as
Anonymous Reviewer (2024) Peer Review #1 of "Single-view 3D reconstruction via dual attention (v0.3)". PeerJ Computer Science

Reviewer 2 ·

Basic reporting

The authors responded to all the comments and suggestions.

Experimental design

The authors responded to all the comments and suggestions.

Validity of the findings

The authors responded to all the comments and suggestions.

Cite this review as
Anonymous Reviewer (2024) Peer Review #2 of "Single-view 3D reconstruction via dual attention (v0.3)". PeerJ Computer Science